# Topology Aware Optimization of Soft Prompts

## Abstract

Soft prompt tuning achieves excellent performance in few-shot tasks. However, soft prompt tuning lacks interpretability, and traditional prompt tuning methods fail to analyze its internal structural features or optimize from this perspective. To address this limitation, this research proposes a topology-aware optimization method focused on the internal structure of soft prompts. By introducing persistent homology methods from topological data analysis (TDA), we characterize the structural evolution features of soft prompts during training, discovering that changes in connectivity, persistence and redundancy affect soft prompt tuning performance. As high-dimensional vectors, soft prompts with stable and concise structures better enhance the performance of large language models (LLMs) on specific tasks. Based on this phenomenon, we developed a new topology aware loss function for optimizing soft prompt training, called TDA for Softprompt Loss Function (TSLoss), which introduces topological measurement tools through TDA to quantify connectivity and redundancy between semantic units, learning information related to topological structure transformations trending toward structural stability. Extensive experiments demonstrate that training with TSLoss can significantly accelerate the convergence speed of prompt tuning while ensuring fine-tuning effectiveness, providing an interpretable research direction for soft prompt tuning from a new perspective.

## 1 Introduction

Prompt Tuning (PT) Lester et al. (2021) is highly parameter-efficient, significantly reducing computational resource consumption and supporting flexible multi-task deployment, making it an economical and effective choice for large model customization and optimization Qin et al. (2021). Soft prompts Vu et al. (2021); Han et al. (2024), which operate in the model's embedding space, require only a few trainable continuous prompt vectors added to the input to achieve effects similar to full parameter fine-tuning. Existing research has demonstrated their superiority through high-quality soft prompts and continues to improve accuracy on specific tasks Vu et al. (2021); Bai et al. (2024).

However, soft prompts consist of trainable continuous vectors that do not correspond to natural language words, making them invisible to humans Schulhoff et al. (2024). This leads traditional methods to focus on accuracy-based evaluation standards to determine if soft prompts successfully guide the LLM's reasoning state. Although current work attempts to analyze soft prompts from an interpretability perspective to enhance generalization Fan et al. (2025), it does not focus on vector analysis in high-dimensional semantic spaces, thus failing to reveal structural differences before and after training and understand the mechanism of "how the model becomes correct."

To address these limitations, we use the persistent homology analysis method Zomorodian & Carlsson (2004) of topological data analysis (TDA) Chazal & Michel (2021) to make the training process of soft prompts interpretable and propose the TDA for Softprompt Loss Function (TSLoss) to improve training accuracy and accelerate convergence. Specifically, this method reveals that during general soft prompt training, the 0-dimensional homology group ($H_0$) remains stable, retaining basic connected components in the reasoning structure; whereas the 1-dimensional homology group ($H_1$) gradually decreases, indicating that loops and redundant structures in the reasoning path reduce and eventually approach zero; persistent homology entropy shows a slight downward trend, indicating that vector structures tend to become simple and stable while maintaining rich information storage. Based on this, we designed TSLoss, combining the characteristics of $H_0$ and $H_1$ in soft prompts training to guide the development of more stable and effective soft prompts for spe-

cific tasks, thereby enhancing representation learning quality. Extensive experiments and theoretical analysis verify the rationality of this phenomenon analysis and the effectiveness of the loss function design. Our contributions are as follows:

- We propose a novel interpretability analysis perspective using persistent homology methods to reveal structural changes in soft prompts during training and their correlation with performance.

- We design a new loss function, TSLoss, specifically for soft prompts, which facilitates convergence and improves their effectiveness and stability.

- Comprehensive experimental validation and theoretical analysis confirm the rationality and effectiveness of our phenomenological analysis and loss fuction design.

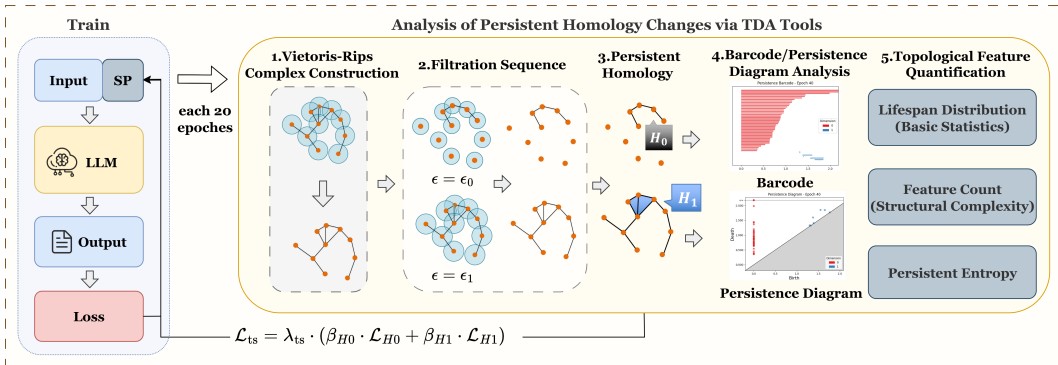

Figure 1: Overview of topological analysis and optimization for soft prompts. This paper first uses persistent homology analysis methods from TDA to reveal inherent phenomena in soft prompts training, and based on these findings, designs an optimization function TSLoss ($L_{ts}$).

## 2 PHENOMENON DEFINITION AND ANALYSIS

In this section, we propose hypotheses from a theoretical perspective for TDA analysis of soft prompt training, and validate and analyze the results using persistent homology.

### 2.1 REDEFINITION OF SOFT PROMPTS

Soft prompts refer to a type of prompt based on neural network models, which guides the model's attention to key information in the input by inserting a specific vector into the input sequence. This helps generate the desired output and can improve the model's performance and adaptability to new tasks through fine-tuning.

At a certain point during the training of a soft prompt, it is typically a learnable parameter matrix $P \in \mathbb{R}^{n \times d}$, where $n$ represents the length of the prompt (number of tokens), and $d$ is the model's hidden dimension (e.g., 768 or 1024).

We can directly treat each row of this matrix as a point in a high-dimensional space, with each point located in a $d$-dimensional embedding space. A soft prompt thus consists of $n$ points, which serve as the initial conditions for analysis using the TDA method during training. From a structural perspective, the training of soft prompts is focused on pattern recognition for a specific task. This structure is simple and stable, as the vector converges to a high-quality output focused on a specific task. Complex structures make it difficult to guide stable outputs for a given task.

### 2.2 ASSUMPTIONS

Treating the soft prompt's token representations as points in a high-dimensional space, under the premise that this metaphor holds, we propose the following hypotheses:

- In LLMs, at the end of each training iteration, each row vector of the soft prompt uniquely determines the position of a point in the $d$-dimensional space. These points have a certain structure in the semantic space and will change during fine-tuning training.
- The structural characteristics of the soft prompt are correlated with its performance. Based on this correlation, methods can be developed to improve both the convergence speed of training and the quality of the generated output.

## 2.3 HYPOTHESIS VERIFICATION THROUGH TDA

Persistent homology Edelsbrunner et al. (2002) is the core method of TDA Wasserman (2018), aimed at capturing topological features (such as connected components and loops) of point cloud data across different scales. The fundamental concept is that as the distance threshold $\epsilon$ increases, the topological structure of the point cloud gradually evolves from isolated points to connected sets, potentially forming higher-dimensional topological features like loops. We choose to quantify the topological changes of soft prompts during training using persistent homology, with the specific analytical notation system shown in Table 1. By continuously tracking these topological features, the analysis captures persistent structures in the data that remain stable across different scales. The core steps of persistent homology in this paper are as follows:

1. **Vietoris-Rips complex construction:** Building complexes based on distances between data points, forming topological representations of point clouds. This is possible as soft prompts form a point set in a metric semantic space with a distance threshold $\epsilon$.

2. **Filtration sequence through scale:** Constructing a series of complexes through changes in parameter $\epsilon$, observing their topological features at different scales.

3. **Homology group calculation:** Computing homology groups at each scale, quantifying topological features, especially connectivity ($H_0$) and redundancy ($H_1$).

4. **Persistence analysis:** Quantifying the persistence of topological features (from appearance to disappearance) through persistence diagrams and barcode plots, calculating measures such as persistence entropy and average lifespan to evaluate structural changes.

Table 1: Notation summary for topological analysis of soft prompts

| Topological Data Analysis Notation | |
| --- | --- |
| Symbol | Description |
| $H_0$ | Zero-dimensional homology group (connected components) |
| $H_1$ | First-dimensional homology group (cycles/loops) |
| $(b_i, d_i)$ | Birth-death pair of $i$-th topological feature |
| $l_i$ | Lifespan of topological feature ($l_i = d_i - b_i$) |
| $L$ | Total lifespan ($L = \sum_i l_i$) |
| PE | Persistence entropy ($-\sum_i \frac{l_i}{L} \log \frac{l_i}{L}$) |
| $\epsilon$ | Filtration parameter (neighborhood radius) |
| $\partial_k$ | Boundary operator of dimension $k$ |

The implementation details of the analysis steps, theoretical design, and experimental results will be described in the appendix. The experimental setup for the analysis can be found in the section 4.

Figure 2 shows an example of four types of topological analysis results on the GSM8K dataset, revealing how soft prompts evolve during normal model training and how inference paths are refined. The (a) average topological feature lifespan curve and (c) topological feature count curve jointly indicate that the number of $H_0$ features remains stable in both single-sample and multi-sample training, suggesting that the model consistently maintains the overall coherence of the inference chain. Meanwhile, the number of $H_1$ features significantly decreases during training, especially faster and more stably in multi-sample training, indicating that the model can utilize structural similarities between samples to quickly identify and eliminate redundant inference paths. This process is verified in (b) persistence diagrams and (d) persistence barcodes. As training progresses, a universal phenomenon emerges where numerous short-lived $H_1$ features gradually disappear, while $H_0$ features become longer and more prominent. The convergence of training is characterized by the formation of a more streamlined and efficient inference structure through the removal of redundant loops.

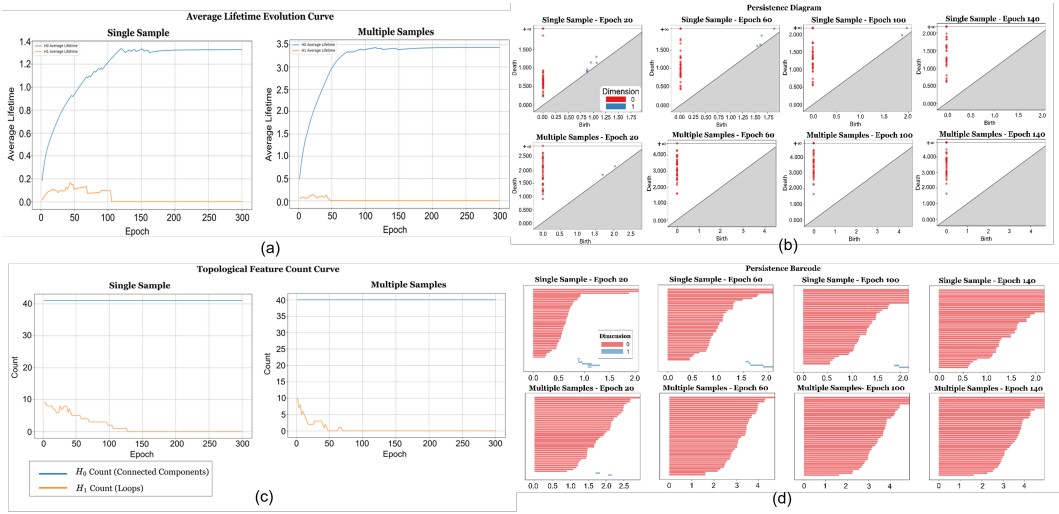

Figure 2: Topological Evolution of Soft Prompts Embedding Space During Training.

Figure 3 analyzes changes in density and topological complexity. The left graph (a) demonstrates that during training, $H_1$ density significantly decreases while the overall density shows only a minimal reduction. This confirms that $H_1$ reduction genuinely results from redundancy elimination rather than from a simple increase in overall point density, as $H_1$ continues to decrease even when density remains relatively constant. The right graph (b) shows how persistence entropy (a metric for topological complexity) rapidly declines from its initial high value before stabilizing. This trend indicates that the model quickly simplifies the topological structure of soft prompts in early training stages by eliminating unnecessary redundancies, creating a more concise and efficient representation. However, the numerical decrease remains modest because while $H_1$ features are reduced, the large number of $H_0$ features persists throughout training. In the left figure (a), the overall point density represents the range and mean values measured after seven iterations.

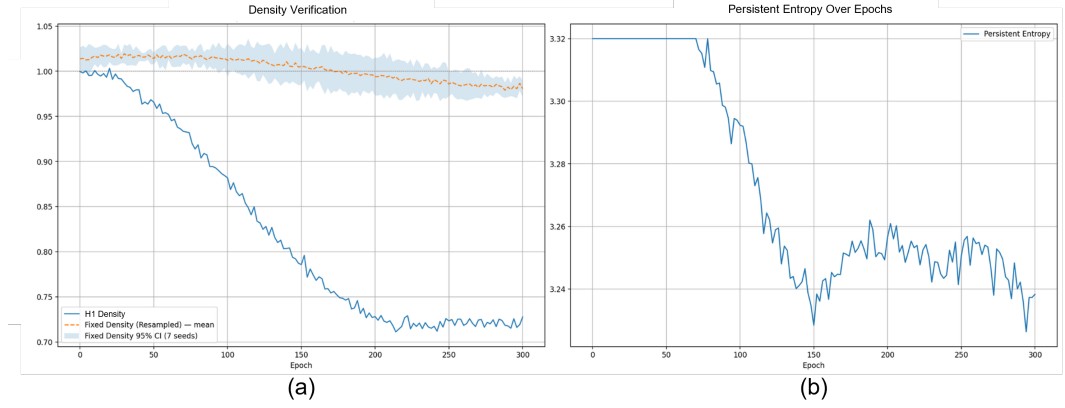

Figure 3: Changes in Density and Topological Complexity of Soft Prompts During Training.

According to Table 2, which presents correlation and significance verification between analytical features and accuracy rates, it can be observed that $H_1$ count shows a certain negative correlation with inference accuracy when applying trained soft prompts to LLMs, indicating that better-performing models have fewer redundant loops. Meanwhile, $H_0$ average lifespan demonstrates an extremely strong positive correlation with accuracy, meaning that more stable inference chains correlate with higher accuracy. Persistence entropy exhibits an extremely strong negative correlation

Table 2: Correlation and Significance of Topological Features and Accuracy

| Metric | Spearman $\rho$ | p-value | Convergence Test |
|---|---|---|---|
| $|H_0|$ (cardinality) | N/A* | N/A | $U = 2664.5, p = 1.000$ |
| $|H_1|$ (cardinality) | $-0.324$ | $6.60 \times 10^{-5}$ | $U = 3600.5, p = 9.7 \times 10^{-5}$ |
| Avg. $l_i$ for $H_0$ | $0.866$ | $3.40 \times 10^{-45}$ | $U = 0.0, p = 1.9 \times 10^{-25}$ |
| Avg. $l_i$ for $H_1$ | $0.018$ | $0.826$ | $U = 2608.0, p = 0.827$ |
| PE | $-0.809$ | $5.10 \times 10^{-35}$ | $U = 5153.0, p = 2.1 \times 10^{-22}$ |

with accuracy, suggesting that more simplified topological structures lead to higher model accuracy. In the validation examples, the $H_0$ count consistently remained a single constant value and was therefore not included in the analysis, therefore, the value is N/A. This analysis confirms that $H_0$ average lifespan and persistence entropy are key indicators for measuring model structure optimization and performance improvement.

## 3 TOPOLOGY AWARE LOSS FUNCTION FOR OPTIMIZING SOFT PROMPTS

Based on TDA analysis results, we design the **TDA for Softprompt Loss Function (TSLoss)**, aimed at controlling the topological structure of data embeddings to ensure the representations learned by the network remain stable and consistent, thereby optimizing the training process. Specifically, according to significance and correlation analysis, TSLoss is designed based on two features: a loss based on H0 features that maintains connectivity by controlling each point's "soft nearest neighbor distance" to regulate local density, and a loss based on H1 features that maintains the local topological structure through enforced attraction and repulsion between points.

Given a set of data points $X = \{x_1, x_2, ..., x_n\}$ in the embedding space, we first define the distance matrix $D_{ij} = \|x_i - x_j\|_2$ for $i, j = 1, \ldots, P$. Since traditional nearest neighbor distance is non-smooth, we designed a Softmin function for gradient computation as

$$s_i = \text{softmin}(D_i, \tau) = -\tau \log \sum_{j=1}^{P} \exp\left(-\frac{D_{ij}}{\tau}\right) \tag{1}$$

where $\tau$ is the temperature parameter of the LLM.

**The $H_0$ feature-based loss** ($\mathcal{L}_{H0}$) stabilizes the connected structure by controlling local density. Note that the soft nearest neighbor distance $s_i$ is a differentiable approximation of the death scale $d_i$ of point $x_i$ in the 0-dimensional homology group in persistent homology. Thus, we design:

$$\bar{s} = \frac{1}{P}\sum_{i=1}^{P} s_i, \quad \mathcal{L}_{H0} = \frac{1}{P}\sum_{i=1}^{P}(s_i - \bar{s})^2 \tag{2}$$

Minimizing $H_0$ is equivalent to forcing all points to have similar local density, equivalent to minimizing the variance of the $H_0$ lifetime distribution, avoiding abnormally sparse or dense regions. This corresponds to the desire for 0-dimensional features to have consistent persistence in the persistent homology diagram, as lifetime and accuracy show strong correlation.

**The $H_1$ feature-based loss** ($\mathcal{L}_{H1}$) focuses on one-dimensional homology features (rings) by balancing attraction and repulsion between point pairs. We prevent unnecessary connectivity through:

$$\mathcal{L}_{\text{repel}} = \frac{1}{n^2}\sum_{i,j} \max(0, \delta - D_{ij})^2 \tag{3}$$

This penalizes point pairs with distances less than $\delta$, preventing the formation of "false" edges at local scale $\delta$, thus avoiding disruption of potential ring structures. And through:

$$\mathcal{L}_{\text{attract}} = \frac{1}{n^2}\sum_{i,j} \max(0, D_{ij} - \zeta)^2 \tag{4}$$

This penalizes point pairs with distances greater than $\zeta$, ensuring appropriate connectivity of data at global scale $\zeta$, preventing ring structure breakage. Combined as:

$$\mathcal{L}_{H1} = \lambda_{\text{repel}} \cdot \mathcal{L}_{\text{repel}} + \lambda_{\text{attract}} \cdot \mathcal{L}_{\text{attract}} \tag{5}$$

This ensures that during the construction of Vietoris-Rips complexes, ring structures form and disappear within appropriate distance threshold ranges.

**Soft quantile threshold calculation** is used to approximate the topological analysis process of TDA across the entire scale range. To determine key distance thresholds, we define soft quantile thresholds:

$$\delta = \sum_{i,j} w_{ij}^{\text{low}} \cdot D_{ij}; \zeta = \sum_{i,j} w_{ij}^{\text{high}} \cdot D_{ij} \tag{6}$$

where $\delta$ is the "soft minimum" of the distance distribution, representing local scale, and $\zeta$ is the "soft maximum" representing global scale. $\alpha$ controls the "sharpness" of the estimate, approaching hard quantile as $\alpha \to \infty$. Meanwhile, the weights are:

$$w_{ij}^{\text{low}} = \frac{e^{-\alpha D_{ij}}}{\sum_{k,l} e^{-\alpha D_{kl}}}; w_{ij}^{\text{high}} = \frac{e^{\alpha D_{ij}}}{\sum_{k,l} e^{\alpha D_{kl}}} \tag{7}$$

Combining the above, the complete topological constancy loss function is:

$$\mathcal{L}_{\text{ts}} = \lambda_{\text{ts}} \cdot (\beta_{H0} \cdot \mathcal{L}_{H0} + \beta_{H1} \cdot \mathcal{L}_{H1}) \tag{8}$$

Where $\lambda_{\text{ts}}$ is the weight of TSLoss. Since this function alone cannot quantify the difference between predicted probability distributions and true label distributions, this loss function can only be used as an auxiliary component alongside other loss functions such as cross-entropy loss Mao et al. (2023).

Under the persistent homology theoretical framework, this loss function combines the conclusions from section 2, by learning the association characteristics of 0-dimensional and 1-dimensional homology groups in soft prompts training, making the trained soft prompts more stable in topological structure, guiding the evolutionary direction of the intrinsic geometric structure, thereby improving the quality of representation learning and the speed of convergence.

## 4 EXPERIMENTS

In this section, we present all detailed settings, models, and datasets used in our validation experiments for topological analysis and loss, demonstrating the effectiveness of TSLoss. Notably, to maintain logical narrative flow and showcase the design rationale for TSLoss, we have chosen to present the topological analysis data in section 2, while providing supplementary details about experimental configurations here.

### 4.1 DATASET AND MODEL SELECTION

To ensure the generality of our evaluation methods, we tested both the topological phenomena analysis and TSLoss optimization effectiveness using three representative benchmark datasets: GSM8K Cobbe et al. (2021) covering elementary mathematics word problems, MMLU-CF Zhao et al. (2024) for common sense and factual questions, and LongBench Bai et al. (2023) for long-context reasoning. These datasets encompass task types ranging from basic calculations to multi-hop reasoning and complex problem solving. We selected two 7B-parameter language models and one 2B-parameter model for TSLoss evaluation: DeepSeek-7B-Chat Bi et al. (2024), Open-LLaMA-7B Geng & Liu (2023), and Gemma-2B-IT, while the topological analysis was only conducted using Gemma-2B-IT. Considering that larger models typically achieve stronger performance through sheer parameter count, where prompt tuning optimizations might be less pronounced and require substantial computational resources, we deliberately chose 7B and 2B models because smaller models respond less stably to prompts compared to larger models. Soft prompts, through learnable contextual shifts, can more effectively activate the model's capabilities on specific tasks while simultaneously conserving computational resources. Additional experimental cases and datasets are presented in the appendix.

## 4.2 IMPLEMENTATION DETAILS

To systematically analyze soft prompt structural changes across different scenarios, we established two experimental configurations: single-sample training and multi-sample training (10 samples, randomly selected). These configurations were used in both our topological phenomenon analysis and TSLoss optimization capability verification. To avoid interference from different data distributions, both training types used samples from the same dataset, ensuring controllability and consistency in comparing structural features.

All soft prompt training processes maintained identical initialization methods, optimizers, and hyperparameters across different task settings. Specifically, soft prompt vectors were initialized using a Gaussian distribution ($\mathcal{N}(0, 0.02^2\mathbf{I})$), with AdamW optimizer and hyperparameters set to learning rate $5 \times 10^{-5}$, $\beta_1 = 0.9$, $\beta_2 = 0.98$, weight decay 0.01, $\epsilon = 10^{-8}$, batch size 8, and 300 training epochs, using a linear learning rate scheduler (with 10% warm-up phase). During training, topological data analysis was performed every 20 epochs while recording inference accuracy to characterize the soft prompt's evolutionary process. Our structural visualization and analysis revealed consistent topological evolution patterns across different datasets and models, demonstrating strong cross-task stability. We present selected samples showing key topological feature changes throughout training, with comprehensive comparisons in the supplementary material. For implementation and analysis, we used a computing environment with an Intel Xeon Platinum 8173M CPU (2.00GHz, 28 cores, 112 threads) and 8 NVIDIA GeForce RTX 3090 GPUs (24GB each).

It is worth noting that in the verification of TSLoss optimization capability, the total loss function used was: $L_{total} = L_{ce} + \lambda_{ts} \cdot L_{ts}$ where $L_{ce}$ represents the cross-entropy loss function.

## 4.3 MAIN RESULT

In the main analysis results, given the specificity of soft prompts for model task fine-tuning, we chose to use soft prompts in single or multiple sample (10 samples) scenarios. In this experiment, we measured the number of training iterations required when these soft prompts were fine-tuned on specific/categorical sample problems to achieve 100% accuracy in LLM problem-solving performance for those sample problems. This approach helps circumvent the difficult-to-measure experimental situations caused by the poor generalization of soft prompts. According to Table 3, adding TSLoss with $\lambda_{ts}$=1 to the cross-entropy loss function improved convergence speed while ensuring correct solutions, achieving good optimization results, with particularly pronounced optimization effects for smaller parameter models.

Table 3: Number of training iterations required for soft prompts to achieve correct problem solutions

| Model | Method | Single-sample | | | Multi-sample | | |
|---|---|---|---|---|---|---|---|
| | | GSM8K | MMLU-CF | LongBench | GSM8K | MMLU-CF | LongBench |
| Open-LLaMA-7B | Standard | 12 | 4 | 6 | 10 | 8 | 38 |
| | +TSLoss | 10 | 2 | 4 | 8 | 4 | 34 |
| DeepSeek-7B-Chat | Standard | 8 | 2 | 4 | 8 | 4 | 32 |
| | +TSLoss | 6 | 2 | 2 | 4 | 2 | 28 |
| Gemma-2B-IT | Standard | 154 | 8 | 12 | 118 | 16 | 104 |
| | +TSLoss | 88 | 6 | 8 | 62 | 14 | 58 |

## 4.4 PARAMETER SENSITIVITY ANALYSIS

Figure 4 shows the convergence of the overall loss function $L_{total}$ under different weight designs. With different $\lambda_{ts}$ weight values, all configurations converge to a level close to zero after approximately 100 epochs, indicating that the topology-aware loss function has good stability across various weight configurations. However, when the $\lambda_{ts}$ weight value is smaller (equal to 1), it facilitates rapid model convergence, but when it becomes too large (equal to 10), the convergence exhibits notable oscillations. This instability contradicts the original design intention, which is also reflected in Table 4. The table shows the specific number of training iterations required during the training process of individual problems, when fine-tuning reaches the point where the LLM inference result is correct for the sample problem. When $\lambda_{ts}$ increases from 0.1 to 1, the number of iterations continuously de-

creases, and convergence becomes faster, while maintaining accuracy. However, when $\lambda_{ts}$ reaches 10, correct convergence cannot be achieved, as the model's focus on structure causes continuous enhancement of initial inference, resulting in no correct rounds being observed.

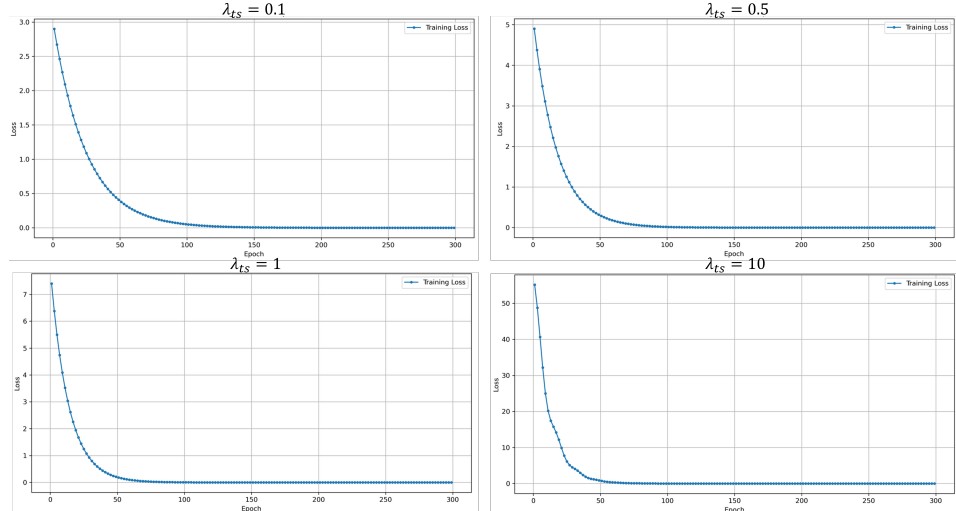

Figure 4: Convergence of overall loss under different TSLoss weight values.

Table 4: Iterations to reach 100% accuracy with varying $\lambda$ values

| $\lambda$ | 0 | 0.1 | 0.5 | 1 | 10 |
|---|---|---|---|---|---|
| Iterations | 154 | 130 | 112 | 88 | – |

## 4.5 GENERALIZATION ANALYSIS

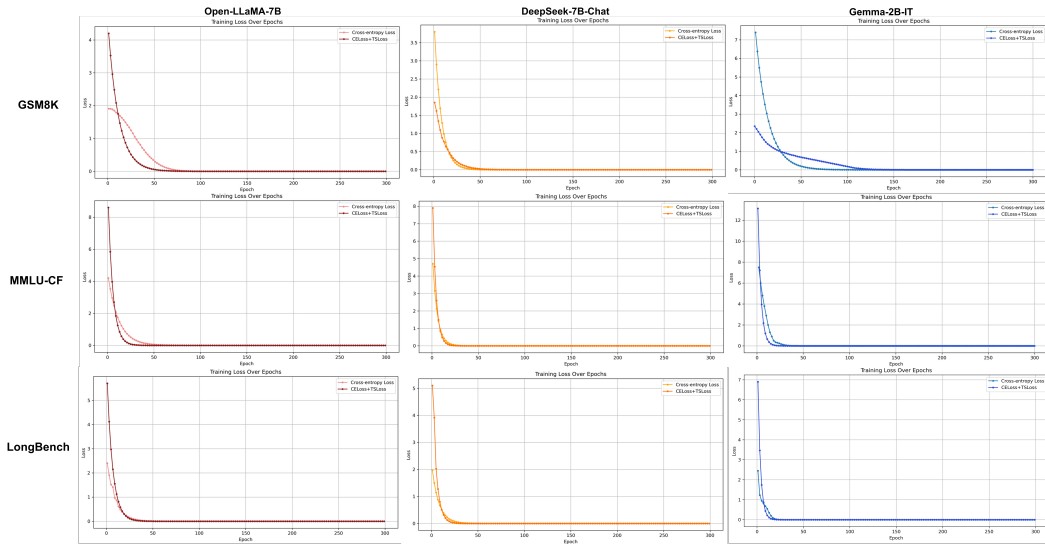

Figure 5: Comparison of convergence behavior of overall loss functions with and without TSLoss across different datasets and models.

In this section, as shown in Figure 5, we compared the performance when using TSLoss + CELoss versus using only the cross-entropy loss function (CELoss) across three datasets and large language

models. The results demonstrate that our optimization method has generalizability across multiple datasets and models, helping soft prompts achieve better convergence effects during fine-tuning.

# 5 RELATED WORK

## 5.1 OVERVIEW OF SOFT PROMPT

Soft prompts evolved from static to dynamic paradigms, enhancing efficiency and adaptability Mangrulkar et al. (2022). Early work focused on static vector optimization, with Prompt Tuning Lester et al. (2021) adding trainable vectors to inputs and Prefix-Tuning Li & Liang (2021) extending parameters across attention layers. Both methods struggled with cross-task generalization and interpretability. The latest research directions include cross-modal extensions and automated design: LASP Bulat & Tzimiropoulos (2022) technology constrains visual-language prompt semantics through text alignment loss; the Automated Prompt Optimization Framework Murthy et al. (2025) achieves "zero-configuration prompt engineering," reducing manual intervention requirements.

Analysis of soft prompts' internal mechanisms employs various specialized approaches. Dynamic intervention techniques like DPC Fan et al. (2025) optimize large language models' reasoning capabilities by selectively suppressing redundant information flow in the embedding space. Recent work has addressed soft prompt interpretability through theoretical frameworks that reveal trade-offs between interpretability and performance Patel et al. (2025). Geometric approaches to prompting have further uncovered distinct representational mechanisms for task adaptation, highlighting how different prompting methods affect representation geometry and the critical role of input distribution samples in few-shot learning contexts Kirsanov et al. (2025). However, existing research lacks comprehensive internal analysis of soft prompts, particularly from a shape characteristic perspective, largely due to the inherent invisibility of these high-dimensional representations.

## 5.2 TDA FOR NATURAL LANGUAGE PROCESS(NLP) ANALYSIS

Topological Data Analysis (TDA) offers a revolutionary approach to understanding complex linguistic structures by capturing the shape and connectivity patterns inherent in language data. Unlike traditional NLP methods that often focus on statistical distributions or contextual similarities, TDA examines the multiscale topological features of data through tools like persistent homology and Mapper algorithm, revealing insights otherwise invisible to conventional techniques Michel et al. (2017). Recent applications have demonstrated TDA's effectiveness across diverse NLP tasks, with researchers leveraging topological signatures to enhance sentiment analysis performance by capturing emotional trajectories in text Gholizadeh et al. (2020), detect AI-generated content through topological inconsistencies in embedding spaces Uchendu et al. (2023), and interpret attention mechanisms in transformer architectures Kushnareva et al. (2021). The topology-preserving properties of TDA make it particularly valuable for analyzing cross-lingual phenomena, as shown by Port et al. Port et al. (2022) who identified invariant syntactic structures across language families that persist despite surface differences. As language models continue to scale, TDA presents promising opportunities for understanding the emergent properties of large language models by analyzing the topological evolution of their representation spaces during training and inference, for example, build a fast and scalable pipeline to characterize the birth and death of topological features across transformer models' layers Gardinazzi et al. (2024).

# 6 CONCLUSION

In this paper, we revealed the topological feature evolution of soft prompts, which are an efficient fine-tuning method for enhancing LLM performance on specialized tasks, and demonstrated the correlation between these features and fine-tuning effectiveness, providing a new perspective for interpretability analysis. We proposed TSLoss, a topology-aware optimization method based on the loss function that effectively improves training convergence speed while ensuring fine-tuning effectiveness. Future work will extend this method to address other inherent problems of soft prompts, such as generalization, and apply it to LLMs with larger parameters.

## 7 REPRODUCIBILITY STATEMENT

To ensure reproducibility of this paper, the authors commit to publishing all implementation details on open-source platforms such as GitHub upon acceptance. Notably, the specific experimental design and configuration details are presented in subsections 4.1 and 4.2, while additional experiments are provided in the appendix, including analyses on other datasets that further confirm the universality of our findings. All reported results represent averages from three independent runs. Additionally, supplementary theoretical validations in the appendix demonstrate the specific implementation steps and theoretical foundations of persistent homology, along with additional details supporting our theoretical validation.

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

# A APPENDIX

## A.1 SUPPLEMENTARY THEORETICAL ANALYSIS

### A.1.1 PERSISTENT HOMOLOGY ANALYSIS METHODS

For the analysis process of soft prompts training, the detailed step-by-step description is as follows:

textbfVietoris-Rips Complex Construction. We first obtain high-dimensional vector representations

$$\mathbf{x}_i$$

for each soft prompt through training, representing the semantic information of each reasoning step. To construct the topological structure of soft prompt vectors, we use the Vietoris-Rips complex, which relies on distance metrics between reasoning steps, specifically the Euclidean distance between vectors. Given

$$X = \{\mathbf{x}_1, \mathbf{x}_2, \ldots, \mathbf{x}_n\}$$

as the set of soft prompt vectors, we define the complex as follows:

$$\text{VR}_\epsilon(X) = \{\sigma \subset X \mid \|\mathbf{x}_i - \mathbf{x}_j\| \leq \epsilon, \forall \mathbf{x}_i, \mathbf{x}_j \in \sigma\} \tag{9}$$

Where

$$\|\mathbf{x}_i - \mathbf{x}_j\|$$

represents the distance between soft prompt vectors

$$\mathbf{x}_i$$

and $\mathbf{x}_j$, and

$$\epsilon$$

is the scale parameter indicating the maximum distance allowed for connection.

**Filtration Sequence Through Scale.** As the scale

$$\epsilon$$

increases, we construct a sequence of nested Vietoris-Rips complexes, forming a filtration

$$\text{VR}_{\epsilon_1}(X) \subset \text{VR}_{\epsilon_2}(X) \subset \cdots \subset \text{VR}_{\epsilon_m}(X)$$

where $\epsilon_1 < \epsilon_2 < \cdots < \epsilon_m$. This allows us to observe how the structure evolves, with richer topological features emerging as

$$\epsilon$$

increases, such as merging of connected components and the formation and disappearance of loops.

**Homology Group Computation.** Once the Vietoris-Rips complex is constructed, we proceed to calculate its homology groups to extract topological features. We primarily focus on two types of homology groups:

The zero-dimensional homology group

$$H_0$$

represents the connectivity of the soft prompt vector set, specifically the number of connected components in the semantic space. A lower

$$H_0$$

value (especially at larger scales) indicates tighter connections between soft prompt vectors, resulting in a more coherent and logically rigorous reasoning chain structure:

$$H_0(X) = \frac{\ker \partial_0}{\text{im} \, \partial_1} \tag{10}$$

The one-dimensional homology group

$$H_1$$

represents the number of loops (i.e., redundancy) in the soft prompt vector set, reflecting potential logical cycles or repetitions in the reasoning paths. A higher

$$H_1$$

value may indicate redundant or repetitive paths in the reasoning chain, increasing the complexity and redundancy of the reasoning process:

$$H_1(X) = \frac{\ker \partial_1}{\operatorname{im} \partial_2} \tag{11}$$

By calculating how homology groups change with scale, specifically through persistence analysis, the stability of soft prompt structural features, including connectivity and redundancy can be quantified. Both

$$H_0$$

and

$$H_1$$

are non-negative integers, with

$$H_0 \geq 1$$

representing connected components and

$$H_1 \geq 0$$

counting loops in the structure.

**Persistence Analysis** visualizes and evaluates the persistence of topological features in soft prompts. Persistence diagrams plot features as points

$$(b, d)$$

representing birth and death scales. Points farther from the diagonal indicate more persistent features, reflecting robust reasoning paths. Barcodes visualize feature lifespans as intervals $[birth, death]$. Longer bars indicate stable structures, while shorter bars may represent transient or redundant elements in reasoning paths. Persistent Entropy quantifies the complexity in feature lifespan distribution:

$$PE = -\sum_i \frac{l_i}{L} \log\left(\frac{l_i}{L}\right) \tag{12}$$

where

$$l_i = death_i - birth_i$$

is each feature's lifespan and

$$L = \sum_i l_i$$

is the total lifespan. Lower values suggest focused, stable reasoning structures; higher values indicate more random structures with potential redundancies. Feature Lifespan Statistics include Maximum Lifespan, $\max(death_i - birth_i)$, representing the most persistent structure's stability, and Average Lifespan, $(1/n) \cdot \sum_i (death_i - birth_i)$, reflecting overall structural stability of reasoning paths.

### A.1.2 CONNECTION BETWEEN 0-DIMENSIONAL HOMOLOGY GROUP VARIANCE AND LIFETIME DISTRIBUTION

The mathematical connection between $H_0$ variance and the 0-dimensional homology group lifetime distribution $l_i$ is based on a key correspondence: the soft nearest neighbor distance $s_i$ is a differentiable approximation of the death scale $d_i$ of point $x_i$ in persistent homology. Since the lifetime of 0-dimensional features equals their death scale (as the birth scale is 0), therefore:

$$\operatorname{Var}(\{s_i\}) \approx \operatorname{Var}(\{d_i\}) = \operatorname{Var}(\{l_i\}) \tag{13}$$

This analogy holds because $s_i = -\tau \log \sum_j \exp(-D_{ij}/\tau)$ is mathematically a smoothed expectation of the distance from point $x_i$ to its nearest neighbor, which precisely determines the scale parameter at which that point merges with other connected components in the persistent homology filtration, i.e., the death scale $d_i$.

### A.1.3 PERSISTENT ENTROPY

Persistent entropy is an information-theoretic measure in topological data analysis that quantifies the uniformity of persistent homology barcode distributions. For a set of persistent homology barcodes $\{(b_i, d_i)\}_{i=1}^n$, where $b_i$ is the birth scale and $d_i$ is the death scale, persistent entropy is defined as:

$$E = -\sum_{i=1}^{n} p_i \log(p_i), \quad \text{where } p_i = \frac{L_i}{L_{\text{total}}}, \quad L_i = d_i - b_i, \quad L_{\text{total}} = \sum_{j=1}^{n} L_j \tag{14}$$

Unlike traditional entropy, in deep learning, maximizing persistent entropy promotes diversity of topological features and avoids domination by a few modes. Therefore, in most cases, lower persistent entropy indicates less stability. However, according to the main text analysis, the training of soft prompts is an optimization for specific tasks, so to meet this condition, the structure should tend toward simplicity and stability while specializing for specific tasks, without losing rich semantic information. Consequently, persistent entropy decreases to some extent, indicating a reduction in redundant structures like $H_1$, but the small magnitude of decrease also suggests that $H_1$ features are fewer and the required semantic information is maintained. Overall, the decrease in persistent entropy is related to improved stability and efficiency of the reasoning chain structure, marking the model's progress toward optimal reasoning paths.

### A.2 ADDITIONAL VISUALIZATION EXPERIMENTS AND ANALYSIS

In this subsection, we provide a comprehensive collection of sample images from additional datasets and models to offer a more complete and intuitive illustration of the soft prompt structure variations. These examples further support the generality and completeness of our analysis across diverse datasets and model architectures.

### A.2.1 ADDITIONAL EXPERIMENTAL SETUPS

The figures presented in this supplementary material are all derived from experiments conducted under the same settings as those in the main paper. They illustrate the structural evolution of soft prompt training across various language models and datasets. We randomly select and show a subset of these figures in the main paper, while the remaining images are provided in full here. All figures are generated based on structural samples collected during the training process, with experimental methods identical to those described in the main paper. The figures are organized by task type (GSM8K, MATH Hendrycks et al. (2021), BBH Suzgun et al. (2022), MMLU-CF, HotpotQA, LongBench Yang et al. (2018)) and training paradigm (single-sample / multi-sample), and within each group, they are arranged in the order of training epochs.

### A.2.2 OVERLAY ANALYSIS OF PERSISTENT HOMOLOGY ($H_0$ AND $H_1$) IN SINGLE-SAMPLE AND MULTI-SAMPLE TRAINING

We conducted a unified averaging analysis of the persistent homology features, specifically $H_0$ and $H_1$, generated during both single-sample and multi-sample training processes. By comparing the persistent homology barcodes and lifetime distributions under different training settings, we observed notable intrinsic regularities in the evolution of reasoning structures, particularly in terms of stability and consistency. These findings not only deepen our understanding of the reasoning chain's evolutionary mechanism from a topological temporal perspective, but also further validate the core conclusions presented in the main text regarding structural convergence and generalization capability. The following figures present the visualization results.

Figure 6 presents the averaged visualization results across experiments on all datasets. The left panel displays results from single-sample visualizations, whereas the right panel shows multi-sample results. Figures 7 and 8 present the visualization results on the GSM8K dataset, with Figure 7 showing single-sample results and Figure 8 showing multi-sample results. Similarly, Figures 9 and 10 correspond to the single-sample and multi-sample results on the MATH dataset; Figures 11 and 12 show results for the BBH dataset; Figures 13 and 14 for the MMLU-CF dataset; Figures 15 and 16 for the HotpotQA dataset; and Figures 17 and 18 for the LongBench dataset. All analyses were still conducted on Gemma-2B-IT.

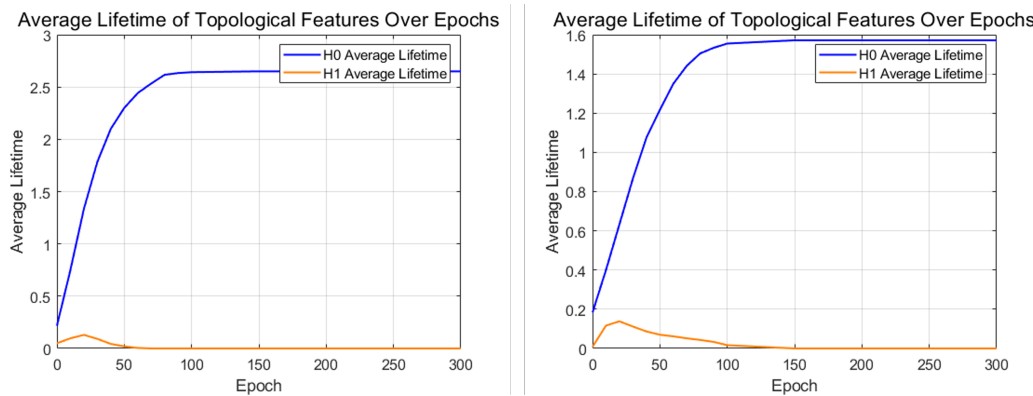

Figure 6: Averaged results of experiments of persistent homology barcodes and lifetime distributions comparison on all datasets.

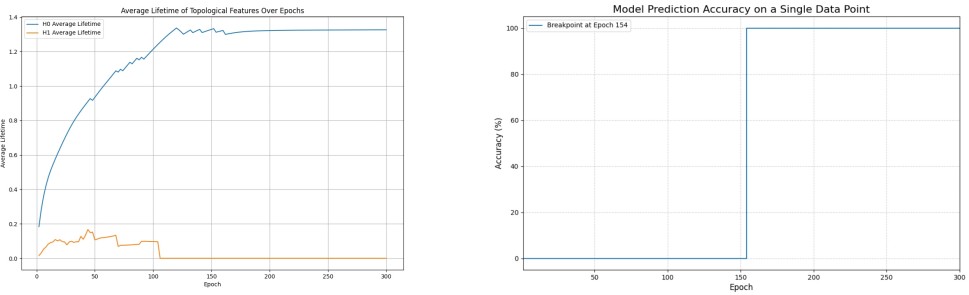

Figure 7: Single-sample results on GSM8K dataset.

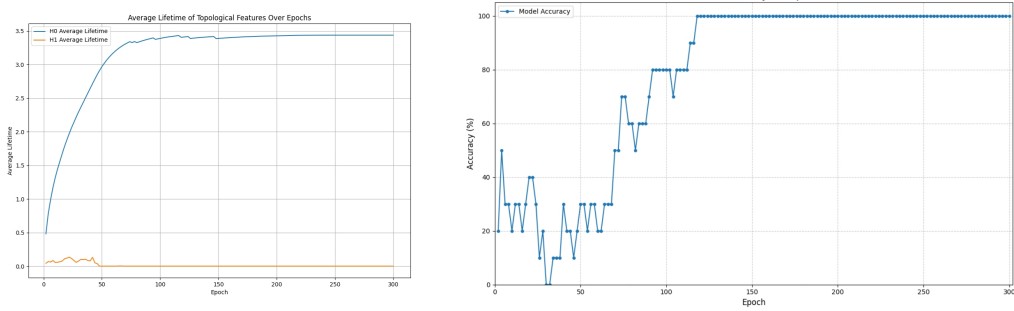

Figure 8: Multi-sample results on GSM8K dataset.

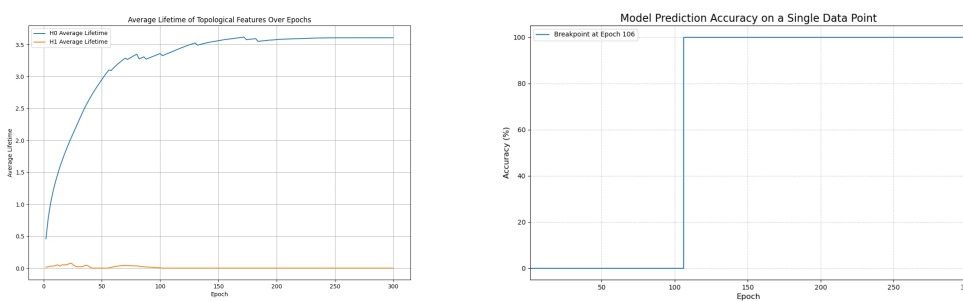

Figure 9: Single-sample results on MATH dataset.

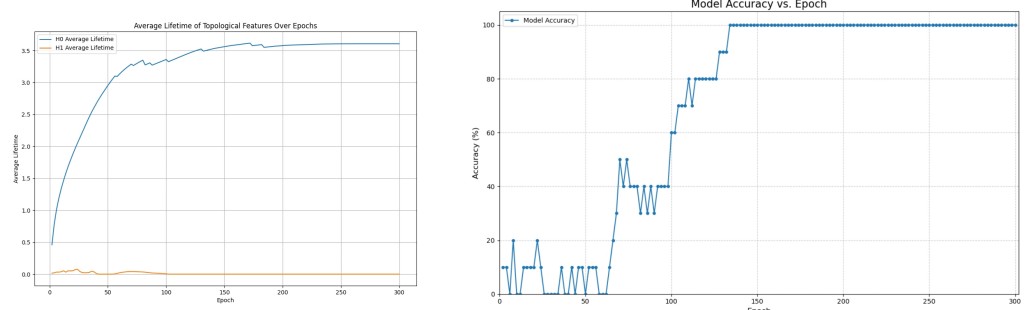

Figure 10: Multi-sample results on MATH dataset.

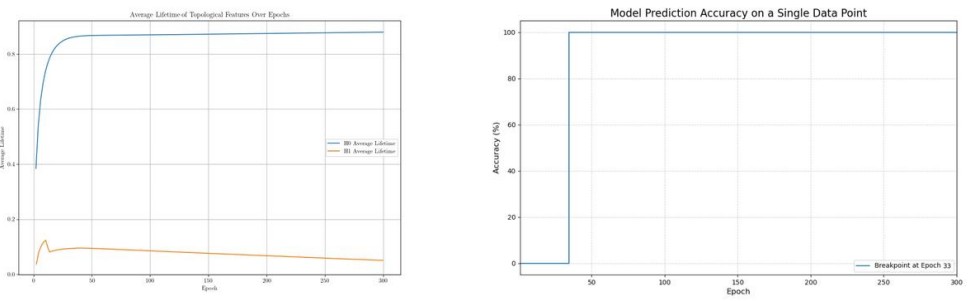

Figure 11: Single-sample results on BBH dataset.

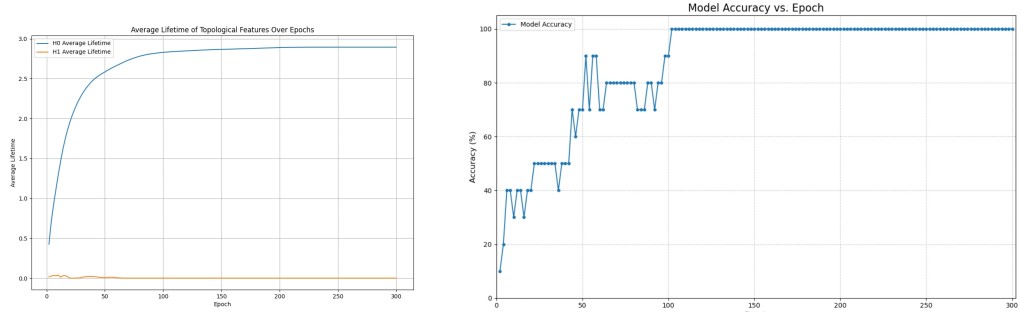

Figure 12: Multi-sample results on BBH dataset.

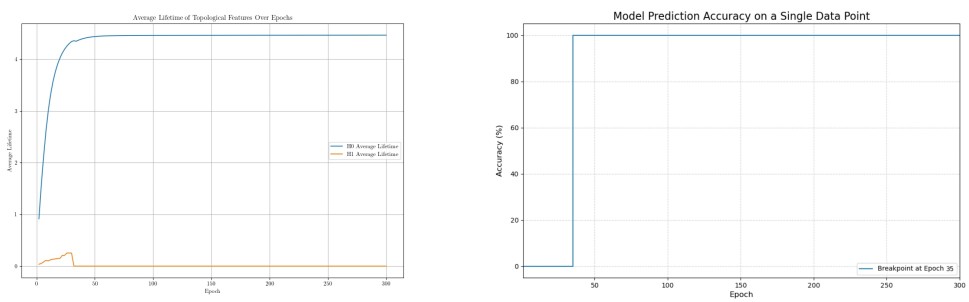

Figure 13: Single-sample results on MMLU-CF dataset.

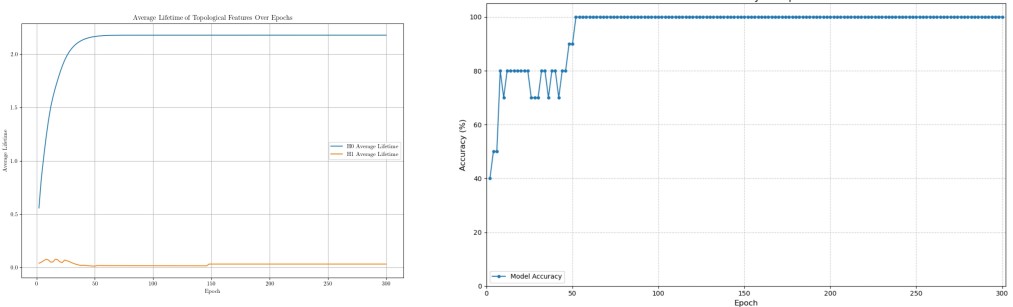

Figure 14: Multi-sample results on MMLU-CF dataset.

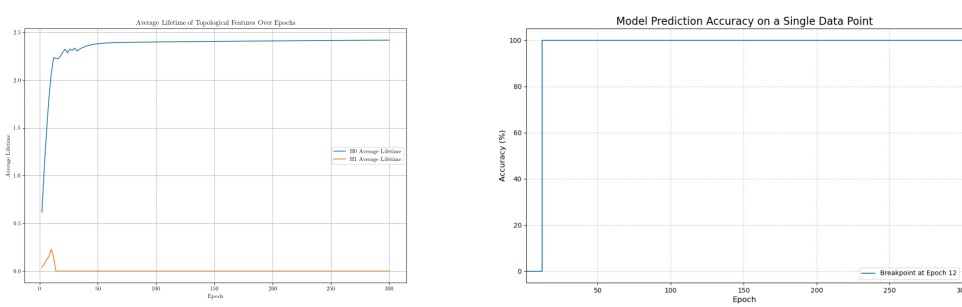

Figure 15: Single-sample results on HotpotQA dataset.

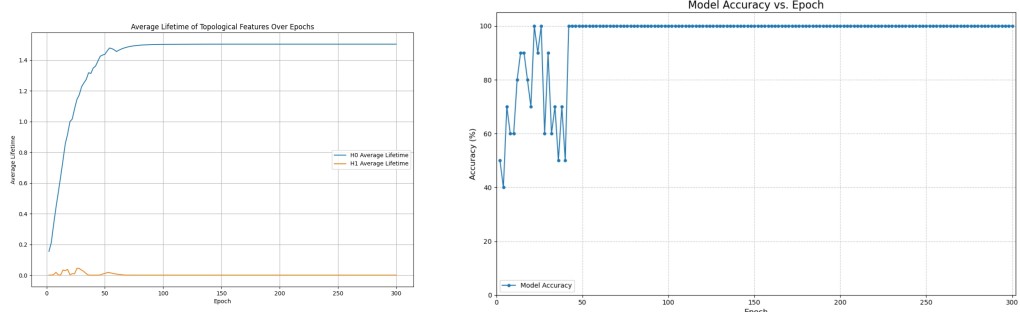

Figure 16: Multi-sample results on HotpotQA dataset.

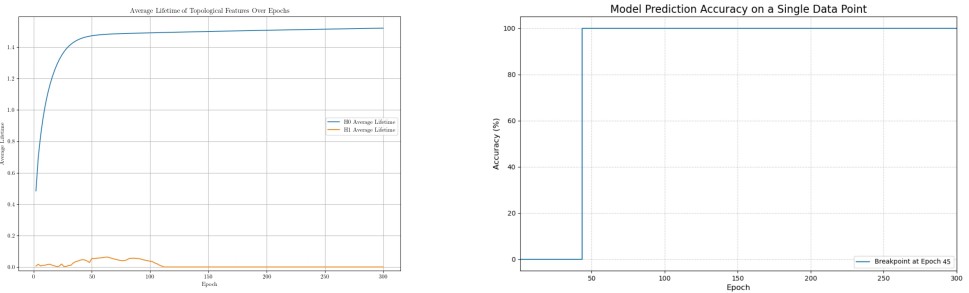

Figure 17: Single-sample results on LongBench dataset.

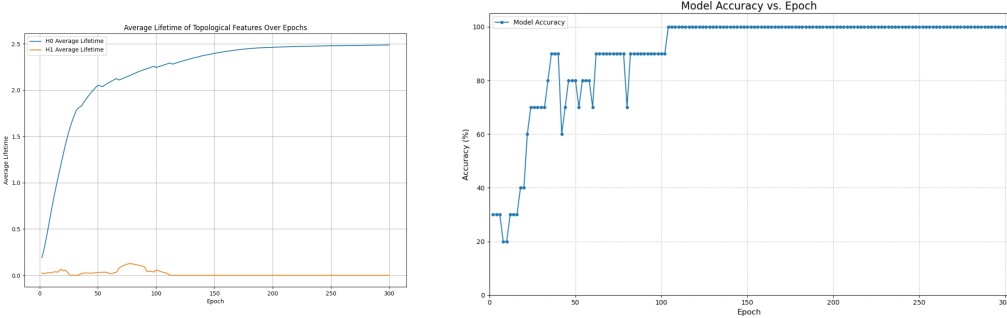

Figure 18: Multi-sample results on LongBench dataset.

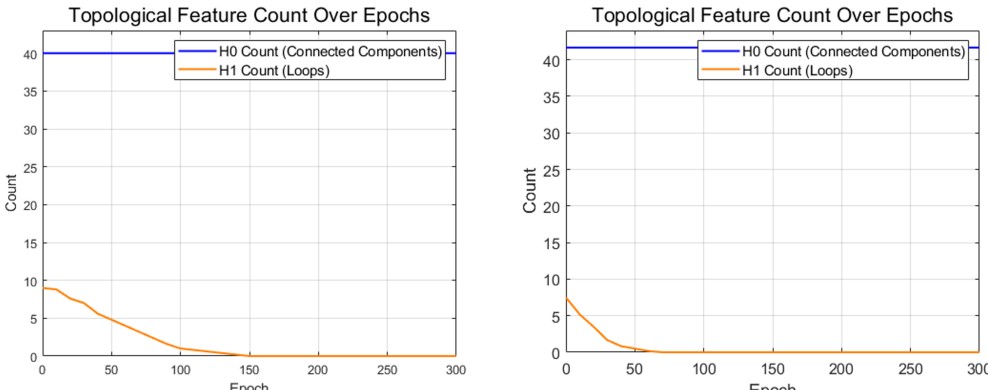

Figure 19: Averaged results of persistent homology in single-sample and multi-sample training on all datasets.

### A.3 Overlay Analysis of the Statistical Variations in $H_0$ and $H_1$ Counts During the Training Process

Following the same methodology used for analyzing the persistent homology ($H_0$ and $H_1$) in single-sample and multi-sample training, we conducted a systematic statistical analysis across six diverse datasets to validate the reliability and generalizability of our conclusions. Specifically, we compared and averaged the quantities of $H_0$ and $H_1$ features and the changes in persistent entropy during both single-sample and multi-sample training processes. Figure 19 shows the averaged results across all six datasets, where the left panel displays results from single-sample visualizations, and the right panel shows multi-sample results. The results in Figure 19 reveal that, despite differences in data distribution and task context, the model exhibits consistent topological evolution trends during the generation of reasoning chains, further reinforcing our core hypothesis regarding structural evolution patterns.

### A.4 Further Discussion

In this paper, we present a novel perspective that reveals the structural vector changes of soft prompts in high-dimensional semantic space during training, providing a new reference for the interpretability of this method to some extent, and design TSLoss based on this phenomenon to optimize the training process. However, our optimization tends to transform soft prompts into more stable and high-quality specialized task fine-tuning tools, which still lacks solutions to the inherent generalization problems of soft prompts design. This will be addressed in our forthcoming work.

Due to computational resource limitations, we focus on smaller parameter models in this paper. However, soft prompt fine-tuning on these capacity-limited models can more significantly enhance task performance and deployment flexibility. Future work will gradually explore the effects on larger parameter models and compare versions of TSLoss adapted for models with different parameter counts.

## A.5 ACKNOWLEDGING THE USE OF LARGE LANGUAGE MODELS (LLMs)

In this paper, we used large language models (such as ChatGPT and Deepseek) as tools for polishing content writing and structural organization. However, they were not used in any way for generating actual academic content or proposing related innovations.

