# OpenReview forum: "Topology aware optimization of soft prompts"
_ICLR.cc/2026/Conference — ICLR 2026 Conference Withdrawn Submission_

### Official Review · Reviewer_X2pG · 2025-10-16

**Soundness:** 2
**Presentation:** 2
**Contribution:** 2
**Rating:** 4
**Confidence:** 3

**Summary:**

The paper introduces a simple topology-aware loss (TSLoss) that stabilizes local density (an (H_0) term via a differentiable soft-nearest-neighbor distance) and modulates pairwise attraction/repulsion using soft quantiles (an (H_1) term), added to cross-entropy. Across GSM8K, MMLU-CF, and LongBench with 2B–7B LLMs, TSLoss consistently reduces the iterations needed to reach correctness, indicating a practical training-efficiency benefit.

**Strengths:**

1. The paper uses persistent homology to track how soft-prompt geometry changes during training and shows concrete correlations—fewer (H_1) loops, larger (H_0) lifetimes, lower persistence entropy—with higher accuracy. That makes the follow-on loss feel motivated rather than retrofitted.

2. TSLoss is a small add-on to cross-entropy, fully differentiable, and easy to implement: softmin for neighbor distance and soft quantiles to set ((\delta,\zeta)). It looks drop-in for existing codebases.

**Weaknesses:**

1. The main reported outcome is “iterations to reach 100% correctness”; it’s **not clearly demonstrated** that TSLoss improves **held-out validation/test performance** under fixed budgets. Reporting standard metrics (EM/F1) on held-out splits would materially strengthen claims.
2. Missing head-to-head against common representational/geometry regularizers under matched budgets.
3. More granular ablations are needed—using only (H_0) vs. only (H_1), alternatives to softmin/soft-quantile, varying temperatures/quantile sharpness, and how each part shifts the observed (H_0/H_1) statistics.
4. The added cost of TSLoss vs. baseline CE is not quantified; scalability beyond 7B and longer prompts is not evaluated.
5. The paper should operationalize what constitutes “useful” transient loops versus “redundant” loops and show evidence that TSLoss suppresses the latter while tolerating the former.

In addition, the resolution of figures other than Figure 1, particularly the font size of text within the figures, needs to be increased.

**Questions:**

Refer to "Weaknesses".

**Details Of Ethics Concerns:**

N/A.

---

### Official Review · Reviewer_Qu19 · 2025-10-27

**Soundness:** 3
**Presentation:** 4
**Contribution:** 3
**Rating:** 8
**Confidence:** 3

**Summary:**

>This paper proposes a topology-aware optimization method for soft prompts, addressing the long-standing issue of lacking interpretability in traditional prompt tuning. By utilizing Persistent Homology analysis from Topological Data Analysis (TDA), the authors were able to characterize the structural evolution of high-dimensional soft prompt vectors during training. The analysis reveals that the stability of the connected components ($H_0$) and the reduction of redundant loops ($H_1$) are strongly correlated with improved performance in Large Language Models (LLMs). Based on these findings, they developed the TDA for Soft-prompt Loss Function (TSLoss), which quantifies connectivity and redundancy to guide the soft prompt toward a more concise and stable topological structure. Extensive experiments demonstrate that training with TSLoss can significantly accelerate convergence speed while maintaining fine-tuning effectiveness across various benchmarks5.

**Strengths:**

>S1. The idea of conducting an internal analysis of soft prompts for the first time is highly appreciated and deemed novel.

>S2. Despite involving complex mathematical concepts, the authors successfully explained the formulas and the qualitative role of each component, resulting in excellent readability.

>S3. The intuition derived from the topological analysis observed across three benchmarks (GSM8K, MMLU-CF, LongBench) provides a strong and well-underpinned foundation for the proposed TS Loss (Topological Structure Loss).

>S4. The design choice to adaptively determine the sensitive parameters ($\delta$ and $\zeta$) used in the $H_1$ loss (repulsion and attraction terms) via soft quantiles is highly practical. This successfully created a loss function that minimizes the need for hyperparameter tuning. Furthermore, the authors clearly rationalized the selection of the adjustable parameter ($\lambda_{ts}$) through a thorough hyperparameter sweep (sensitivity analysis).

**Weaknesses:**

This paper is well-executed, and I have only minor suggestions for improvements and further investigation.

>W1. To enhance the understanding of the TDA analysis results, providing a qualitative visualization would be beneficial. For instance, demonstrating how the soft prompt data points (represented in $d$-dimensions) evolve over training using a dimensionality reduction technique such as PCA or t-SNE could offer valuable insights.

>W2. While the main manuscript generally explains the core topology concepts well, some concepts requiring specific prior knowledge, such as the persistence diagram and barcode, lack sufficient qualitative explanation. For readers unfamiliar with Topological Data Analysis (TDA), this necessitated external research, impacting the initial time required for comprehension.

>W3. Some text elements within the figures are too tiny to be easily read, hindering the interpretation of visual evidence.

**Questions:**

>The proposed TS Loss seems to maintain the local density and local topological structure of the soft prompts through attraction and repulsion mechanisms. Could the authors provide a qualitative description of the resulting alignment of the soft prompt data points? For example, does minimizing the TS Loss encourage the data points to be organized into a grid-like structure, or some other specific geometric shape, in the $d$-dimensional space?

---

### Official Review · Reviewer_4vMW · 2025-10-28

**Soundness:** 2
**Presentation:** 1
**Contribution:** 3
**Rating:** 2
**Confidence:** 4

**Summary:**

This paper introduces a topology-aware optimization framework for soft prompt tuning in large language models (LLMs), leveraging persistent homology from Topological Data Analysis (TDA) to interpret and improve the internal structure of soft prompts during training. The authors observe that high-performing soft prompts exhibit stable 0-dimensional homology (H0, representing connectivity) and diminishing 1-dimensional homology (H1, representing loops/redundancy). Based on these empirical findings, they propose TSLoss—a novel auxiliary loss function that regularizes the geometric structure of soft prompt embeddings by encouraging uniform local density (via H0-inspired terms) and controlling inter-point distances to suppress spurious loops (via H1-inspired attraction/repulsion terms). Experiments on multiple datasets (GSM8K, MMLU-CF, LongBench) and models (Gemma-2B, Open-LLaMA-7B, DeepSeek-7B) show that TSLoss accelerates convergence and improves fine-tuning effectiveness, especially for smaller models. The work bridges geometric interpretability and parameter-efficient tuning, offering a new lens to understand how soft prompts evolve during training.

**Strengths:**

(1) The integration of persistent homology from TDA into soft prompt analysis is highly innovative, providing a mathematically grounded and interpretable perspective on the otherwise opaque structure of continuous prompts.
(2) The empirical observations linking H0 lifespan, H1 count, and persistence entropy to model accuracy are insightful and consistently validated across datasets and training regimes (single vs. multi-sample).
(3) The proposed TSLoss is well-motivated by topological phenomena and carefully designed with differentiable approximations (e.g., softmin for nearest-neighbor distances, soft quantiles for scale thresholds), enabling practical integration into standard training pipelines.
(4) The experimental design is thorough, including ablation on loss weights, convergence comparisons, and cross-model/dataset generalization tests, demonstrating robustness and practical utility of the method.

**Weaknesses:**

(1) It seems that the paper lacks the basic report of performance score for proposed methods as well as the compared methods. Also,
the paper claims that H0 cardinality remains constant (a single connected component) across training, justifying its exclusion from correlation analysis (Table 2). However, this may not hold universally—especially in multi-sample or multi-task settings where prompts could fragment. Can the authors clarify under what conditions H0 count stays constant, and provide evidence (e.g., variance across runs or tasks) that this isn’t an artifact of the specific initialization or dataset choice?
(2) TSLoss uses soft quantile thresholds δ and ζ derived from the full distance matrix to define local and global scales. However, in practice, the softmin and softmax weighting (Eq. 7) may be dominated by extreme distances, especially in high-dimensional spaces where distance concentration occurs. How sensitive is TSLoss to the choice of temperature-like parameter α in Eq. 7, and was α tuned per dataset or fixed? A sensitivity analysis for α would strengthen confidence in the method’s stability.
(3) The experiments focus on small models (2B–7B parameters) and few-shot (1 or 10 samples) settings. While the authors justify this by noting larger models are less prompt-sensitive, it remains unclear whether TSLoss provides benefits in more realistic few-shot scenarios (e.g., 16–32 examples) or with stronger base models (e.g., Llama-3-8B). Could the authors comment on scalability and whether the topological phenomena (e.g., H1 decay) still hold in such regimes?
(4) The loss function combines LH0 and LH1 with fixed coefficients βH0 and βH1 (Eq. 8), but their values are not specified in Section 4.2. Were these set to 1, or tuned? If tuned, how? If fixed, what is the justification? Moreover, since LH0 minimizes variance of soft nearest-neighbor distances, could this inadvertently collapse all prompt tokens into a single point, harming expressivity?
(5) The paper states that TSLoss is used alongside cross-entropy loss (Ltotal = Lce + λts·Lts), but it’s unclear how gradients from TSLoss interact with those from Lce. Does TSLoss dominate early training (when structure is chaotic) and fade later? Or does it consistently influence optimization? A plot of the relative magnitudes of ∇Lce and ∇Lts over epochs would help understand the dynamics.
(6) The correlation analysis in Table 2 uses Spearman ρ, which measures monotonic relationships, but the actual relationship between topological features and accuracy may be non-monotonic (e.g., optimal H1 count might be non-zero). Could the authors provide scatter plots or conditional accuracy curves (e.g., accuracy vs. PE bins) to verify the nature of these dependencies?
(7) The method assumes soft prompt tokens form a point cloud in Euclidean embedding space, but LLM embeddings often lie on or near a nonlinear manifold. Does using Euclidean distance in Vietoris-Rips complexes misrepresent true semantic relationships? Have the authors considered alternative metrics (e.g., cosine similarity) or manifold-aware complexes?
(8) In multi-sample training, are the soft prompts shared across samples or sample-specific? If shared (as is typical), the point cloud comprises n×k points (k samples), which may conflate intra- and inter-sample structure. How does the TDA analysis disentangle task-relevant topology from sample-specific noise? Clarifying the prompt architecture in multi-sample settings is essential for reproducibility.

**Questions:**

(1) It seems that the paper lacks the basic report of performance score for proposed methods as well as the compared methods. Also,
the paper claims that H0 cardinality remains constant (a single connected component) across training, justifying its exclusion from correlation analysis (Table 2). However, this may not hold universally—especially in multi-sample or multi-task settings where prompts could fragment. Can the authors clarify under what conditions H0 count stays constant, and provide evidence (e.g., variance across runs or tasks) that this isn’t an artifact of the specific initialization or dataset choice?
(2) TSLoss uses soft quantile thresholds δ and ζ derived from the full distance matrix to define local and global scales. However, in practice, the softmin and softmax weighting (Eq. 7) may be dominated by extreme distances, especially in high-dimensional spaces where distance concentration occurs. How sensitive is TSLoss to the choice of temperature-like parameter α in Eq. 7, and was α tuned per dataset or fixed? A sensitivity analysis for α would strengthen confidence in the method’s stability.
(3) The experiments focus on small models (2B–7B parameters) and few-shot (1 or 10 samples) settings. While the authors justify this by noting larger models are less prompt-sensitive, it remains unclear whether TSLoss provides benefits in more realistic few-shot scenarios (e.g., 16–32 examples) or with stronger base models (e.g., Llama-3-8B). Could the authors comment on scalability and whether the topological phenomena (e.g., H1 decay) still hold in such regimes?
(4) The loss function combines LH0 and LH1 with fixed coefficients βH0 and βH1 (Eq. 8), but their values are not specified in Section 4.2. Were these set to 1, or tuned? If tuned, how? If fixed, what is the justification? Moreover, since LH0 minimizes variance of soft nearest-neighbor distances, could this inadvertently collapse all prompt tokens into a single point, harming expressivity?
(5) The paper states that TSLoss is used alongside cross-entropy loss (Ltotal = Lce + λts·Lts), but it’s unclear how gradients from TSLoss interact with those from Lce. Does TSLoss dominate early training (when structure is chaotic) and fade later? Or does it consistently influence optimization? A plot of the relative magnitudes of ∇Lce and ∇Lts over epochs would help understand the dynamics.
(6) The correlation analysis in Table 2 uses Spearman ρ, which measures monotonic relationships, but the actual relationship between topological features and accuracy may be non-monotonic (e.g., optimal H1 count might be non-zero). Could the authors provide scatter plots or conditional accuracy curves (e.g., accuracy vs. PE bins) to verify the nature of these dependencies?
(7) The method assumes soft prompt tokens form a point cloud in Euclidean embedding space, but LLM embeddings often lie on or near a nonlinear manifold. Does using Euclidean distance in Vietoris-Rips complexes misrepresent true semantic relationships? Have the authors considered alternative metrics (e.g., cosine similarity) or manifold-aware complexes?
(8) In multi-sample training, are the soft prompts shared across samples or sample-specific? If shared (as is typical), the point cloud comprises n×k points (k samples), which may conflate intra- and inter-sample structure. How does the TDA analysis disentangle task-relevant topology from sample-specific noise? Clarifying the prompt architecture in multi-sample settings is essential for reproducibility.

---

### Official Review · Reviewer_6p7L · 2025-11-01

**Soundness:** 2
**Presentation:** 3
**Contribution:** 2
**Rating:** 2
**Confidence:** 3

**Summary:**

This paper introduces a novel approach to optimizing soft prompt tuning in large language models (LLMs) through topological data analysis (TDA). The authors identify a critical gap in current soft prompt research: the lack of interpretability regarding the internal structural features of soft prompts during training. To address this, they apply persistent homology methods from TDA to analyze how soft prompts evolve topologically during training. Their key finding is that certain topological properties (connectivity measured by H0, redundancy measured by H1, and persistence entropy) correlate with tuning performance. Based on these observations, they propose TSLoss, a topology-aware loss function that guides soft prompt training toward more stable and concise structures. Extensive experiments across multiple datasets (GSM8K, MMLU-CF, LongBench) and models (DeepSeek-7B-Chat, Open-LLaMA-7B, Gemma-2B-IT) demonstrate that TSLoss accelerates convergence while maintaining fine-tuning effectiveness.

**Strengths:**

- The paper builds on solid theoretical foundations from TDA (persistent homology) and provides clear mathematical connections between topological concepts and soft prompt properties.

- The authors convincingly identify consistent topological patterns during soft prompt training (stable H0, decreasing H1, decreasing persistence entropy) across multiple datasets and models, with strong statistical correlations to performance (Table 2).

- TSLoss appears to deliver tangible benefits, reducing training iterations by 20-40% across various models and datasets while maintaining accuracy. The parameter sensitivity analysis provides useful implementation guidance.

**Weaknesses:**

1. The primary evaluation metric—number of iterations to achieve 100% accuracy on specific training samples—fails to address the critical generalization problem of soft prompts. The paper doesn't evaluate performance on unseen examples, which is the primary challenge in prompt engineering.

2. There's no comparison with other state-of-the-art prompt optimization methods (e.g., P-tuning, Prefix-tuning, or more recent approaches). This makes it difficult to assess the relative value of TSLoss compared to existing alternatives.

3. While the paper establishes strong correlations between topological features and performance (Table 2), it doesn't convincingly demonstrate causation. The theoretical justification for why these specific topological properties affect performance remains somewhat speculative.

4. The paper mentions performing TDA analysis every 20 epochs but doesn't quantify the additional computational cost or memory requirements of TSLoss compared to standard soft prompt tuning.

5. The method appears highly sensitive to the λ_ts parameter (performance degrades significantly at λ_ts=10), yet the paper provides limited guidance on how to select this parameter for different tasks or models.

**Questions:**

The causal relationship between topological properties and performance isn't fully established—while Table 2 shows strong correlations, the paper doesn't rule out the possibility that these topological features are merely coincidental byproducts of effective training rather than causal factors.

The evaluation metrics are significantly limited: focusing exclusively on convergence speed to 100% accuracy on training samples ignores the critical issue of generalization to unseen examples. There's no ablation study showing the individual contribution of H₀ and H₁ loss components, which would help understand which topological feature is more impactful. The parameter sensitivity analysis (Section 4.4) is good but limited to a single model/dataset combination.

Some TDA-specific terminology (e.g., "persistence entropy," "Vietoris-Rips complex") could be better explained for readers unfamiliar with topological data analysis. Some figures (particularly Figure 2) would benefit from more detailed captions explaining what specific patterns readers should focus on. More explanations are suggested to highlight the advantages of applying the designed losses in prompt tuning.

Fore related works, there could be deeper engagement with recent geometric approaches to prompting (Kirsanov et al., 2025 is mentioned but not analyzed in depth). The paper could better position itself against other interpretability-focused prompt tuning methods like Patel et al. (2025), which directly addresses "towards interpretable soft prompts."

---

### Official Review · Reviewer_JS2Q · 2025-11-01

**Soundness:** 2
**Presentation:** 2
**Contribution:** 2
**Rating:** 4
**Confidence:** 2

**Summary:**

The paper proposes Topology-Aware Optimization of Soft Prompts, introducing a topology-aware loss function (TSLoss) based on Topological Data Analysis (TDA). By analyzing the structural evolution of soft prompt embeddings through persistent homology, the authors identify correlations between topological features (H₀, H₁, and persistence entropy) and model performance. TSLoss leverages these findings to promote stable and concise prompt structures, leading to faster convergence across datasets such as GSM8K, MMLU-CF, and LongBench using models like Gemma-2B, Open-LLaMA-7B, and DeepSeek-7B.

**Strengths:**

1. The idea of using TDA to analyze the internal geometry of soft prompts sounds novel.

**Weaknesses:**

1. The experimental evaluation is limited and lacks strong baselines. Table 3 compares the proposed TSLoss only with standard cross-entropy training, omitting other established soft-prompt optimization or parameter-efficient tuning methods (e.g., Prefix-Tuning, P-Tuning v2, LoRA, LASP, DPC, or sparsity-based prompt pruning). Without these comparisons, it is unclear whether the observed gains in convergence speed stem from the proposed topology-aware loss or simply from general regularization effects. This weakens the empirical validity and makes it difficult to assess the true advantage of the method.

2. Although TSLoss requires fewer iterations to reach correct solutions, the computational cost per iteration may be higher due to the additional topological regularization. This trade-off between convergence speed and per-iteration overhead should be discussed and quantified.

**Questions:**

N/A

---

### Note · Authors · 2025-11-15

I have read and agree with the venue's withdrawal policy on behalf of myself and my co-authors.